# MRI Radiomics Data Analysis for Differentiation between Malignant Mixed Müllerian Tumors and Endometrial Carcinoma

**DOI:** 10.3390/cancers16152647

**Published:** 2024-07-25

**Authors:** Mayur Virarkar, Taher Daoud, Jia Sun, Matthew Montanarella, Manuel Menendez-Santos, Hagar Mahmoud, Mohammed Saleh, Priya Bhosale

**Affiliations:** 1Department of Radiology, University of Florida College of Medicine-Jacksonville, Jacksonville, FL 32209, USA; mayur.virarkar@jax.ufl.edu (M.V.); matthew.montanarella@jax.ufl.edu (M.M.); 2Department of Radiology, University of Texas MD Anderson Cancer Center, Houston, TX 77030, USA; tedaoud@mdanderson.org (T.D.); jsun9@mdanderson.org (J.S.); hag.smahmoud@gmail.com (H.M.); mosaleh@ucsd.edu (M.S.); priya.bhosale@mdanderson.org (P.B.)

**Keywords:** endometrial cancer, artificial intelligence, machine learning, mixed Müllerian tumor, radiomics

## Abstract

**Simple Summary:**

Endometrial cancer and mixed Müllerian tumors are different types of uterine malignancies, yet differentiating these diseases can be challenging and often requires surgery. A timely and accurate diagnosis of these conditions is crucial for providing optimal treatment options. The emerging field of radiomics may assist clinicians in diagnosing these conditions preoperatively by extracting complex patterns from clinical imaging to identify features unique to each type of tumor. This study aims to analyze radiomics data from patients with endometrial cancer and mixed Müllerian tumors to identify distinguishing features that could help clinicians diagnose each disease.

**Abstract:**

The objective of this study was to compare the quantitative radiomics data between malignant mixed Müllerian tumors (MMMTs) and endometrial carcinoma (EC) and identify texture features associated with overall survival (OS). This study included 61 patients (36 with EC and 25 with MMMTs) and analyzed various radiomic features and gray-level co-occurrence matrix (GLCM) features. These variables and patient clinicopathologic characteristics were compared between EC and MMMTs using the Wilcoxon Rank sum and Fisher’s exact test. The area under the curve of the receiving operating characteristics (AUC ROC) was calculated for univariate analysis in predicting EC status. Logistic regression with elastic net regularization was performed for texture feature selection. This study showed that skewness (*p* = 0.045) and tumor volume (*p* = 0.007) significantly differed between EC and MMMTs. The range of cluster shade, the angular variance of cluster shade, and the range of the sum of squares variance were significant predictors of EC status (*p* ≤ 0.05). The regularized Cox regression analysis identified the “256 Angular Variance of Energy” texture feature as significantly associated with OS independently of the EC/MMMT grouping (*p* = 0.004). The volume and texture features of the tumor region may help distinguish between EC and MMMTs and predict patient outcomes.

## 1. Introduction

Endometrial carcinoma (EC) and malignant mixed Müllerian tumors (MMMTs), also known as carcinosarcomas, represent two distinct yet interrelated entities within the spectrum of uterine malignancies. EC, characterized by the abnormal growth of endometrial cells, comprises a heterogeneous group of tumors with varying histological subtypes and clinical behaviors [1]. In contrast, an MMMT is a rare but aggressive tumor composed of both epithelial and mesenchymal components, posing challenges in accurate diagnosis and management [2].

The differentiation between EC and MMMTs holds paramount clinical significance due to their divergent treatment approaches, prognostic implications, and therapeutic outcomes [3]. While EC is often associated with favorable survival rates, MMMT is ominous for its aggressive behavior, leading to poorer clinical outcomes. However, distinguishing between these two malignancies based solely on clinical and histological grounds can be complex, necessitating advanced diagnostic tools for accurate classification [4].

In recent years, radiomic analysis has steered into a new era of precision medicine in oncology. Radiomics analysis has emerged as a promising tool for the non-invasive characterization of tumors and the prediction of clinical outcomes and treatment responses in various cancer types [5,6,7]. It accomplishes this by identifying and analyzing very subtle patterns in the area of interest of the imaging study. In our study, the area of interest is the uterine tumor. Varying patterns include tumor texture, intensity, size, and shape [8]. This approach holds immense promise in unraveling intricate gray-scale patterns within images that might otherwise escape the human eye. In gynecological malignancies, including EC and MMMTs, radiomics, though still in infancy, has emerged as a potential avenue to enhance diagnostic accuracy, prognostication, and treatment stratification.

While existing studies have begun to explore the utility of radiomics in differentiating EC from MMMTs, significant knowledge gaps persist [4]. In particular, the precise radiomic features that delineate these malignancies, the potential correlation of these features with clinicopathological characteristics, and the integration of radiomics into clinical decision-making frameworks remain areas of active research. Furthermore, introducing novel machine learning algorithms has opened avenues for developing predictive models that could revolutionize the diagnostic landscape of EC and MMMTs [9].

This study presents a comprehensive analysis of radiomic features extracted from MRI images of patients diagnosed with EC and MMMTs. We aim to elucidate the specific radiomic features that could differentiate these malignancies.

## 2. Materials and Methods

This retrospective study was performed at our institution after the approval of the institutional review board, which granted a waiver of informed consent.

### 2.1. Patient Population

The institution’s database was used to identify patients with histologically proven uterine MMMTs or EC. A computerized search for “carcinosarcoma” and “mixed Mullerian tumor” was performed using the institution’s database for cases and yielded 1364 patients with MMMTs treated from January 2002 to December 2015. A computerized search for “endometrial adenocarcinoma” and “endometrial carcinoma” for cases yielded 154 patients with EC treated from December 2013 to December 2015.

#### 2.1.1. Inclusion Criteria

From the total query results, patients were included in the study if they met the following criteria: (1) a diagnosis of MMMT or EC; (2) a staging MRI performed prior to chemotherapy, radiation, or surgery; (3) visible tumor on MRI; and (4) available tumor pathology report from our institution. Of the 1518 patients found in the initial computerized search, 61 matched the inclusion criteria and were included in the study. Of these, 42 patients had a diagnosis of MMMT, and 25 patients had a diagnosis of EC.

#### 2.1.2. Exclusion Criteria

Patients were excluded from the study that met any of the following criteria: (1) staging MRI performed after chemotherapy, radiation, or surgery; (2) no visible tumor on staging MRI; or (3) staging MRI with motion degradation.

### 2.2. MRI Techniques

MRI studies were performed at our institution on 1.5 T machines (GE Signa HDxt, manufactured in Florence, SC, USA) using an eight-channel cardiac or multi-channel body coil. To limit artifacts caused by bowel peristalsis, the patients were instructed to fast for 4–6 h before the scan. Dynamic contrast-enhanced (DCE) images were acquired using T1 DCE which was performed using a gradient echo sequence with the following parameters: TR (repetition time) 4–6 ms, TE (echo time) 1–2 ms, field of view 220–240 mm, flip angle 12–15°, matrix 150–208 × 256–320, section thickness 2.5–5.0 mm, interslice gap 2.0–2.5 mm, and excitation. MRI studies for 14 cases (13 MMMT cases and 1 EC case) were conducted at an outside facility (OSF).

### 2.3. Tumor Segmentation and Radiomic Feature Extraction

Axial T2 FSE nonfat saturated images (small FOV–TR/TE) were used for segmentation, and sagittal T2 images were used to calculate the tumor volume. The endometrial primary tumors were manually contoured using the free, open-source software application 3D Slicer 5.2 (https://www.slicer.org/ (accessed on 4 April 2020)) (Figure 1). The segmentations were conducted by a post-doc research assistant and confirmed with a radiologist (PB, with 18 years of experience in pelvic MRI reading) blinded to clinical and pathological patient information. Although not used for segmentation, the sagittal dynamic postcontrast T1 and DWI and sagittal T2 sequences were available to the readers for visual inspection to verify tumor borders. The radiomic tumor features extracted from the masks were 300 first- and second-order radiomic features. The radiomic features were computed by 3D Slicer 5.2. To compare whole-tumor radiomic profiling with single-slice radiomic profiling, the axial image planes depicting the largest tumor mask area were identified, and all the radiomic features were also extracted from these single-slice masks.

### 2.4. Statistical Analysis

Patient characteristics were summarized using 300 features of first-order statistics: average and range measures, angular second-order statistics: angular variance measures, Different Scale Levels: 8 × 8 window, 16 × 16 window, 32 × 32 window, 64 × 64 window, 256 × 256 window. Additionally, tumor volume was also analyzed. These variables were compared between endometrial carcinoma and malignant mixed Müllerian tumors using the Wilcoxon Rank sum and Fisher’s exact test. Area Under the Curve of the Receiver Operating Characteristics (AUC ROC) was calculated for univariate analysis to predict EC status. Logistic regression with elastic net regularization was performed for texture feature selection. The elastic net is a regularized regression method that linearly combines the penalties of the lasso and ridge methods. Both regularization and mixing parameters were optimized using 5-fold cross-validation based on the AUC ROC. Overall survival (OS) curves were estimated using the Kaplan–Meier method. A log-rank test was used to compare survival curves. Regularized Cox regression with elastic net was used to select OS texture features. The regularization parameter was optimized using 5-fold cross-validation based on the Harrell C index. A *p*-value less than 0.05 was considered statistically significant. Statistical analyses were conducted using R (version 3.63, R Development Core Team, Vienna, Austria).

## 3. Results

### 3.1. Patient Characteristics

A total of 61 patients were included in the study: 36 patients with EC and 25 with MMMT. The patients with MMMT had a mean age of 66.5 years (range 24–83 years), while the mean age of the patients with EC was 64.3 years (range 30–80 years). A higher proportion of Hispanic patients existed in the EC group (9%) than in the MMMT group (3%), whereas the proportion of black patients was higher in the MMMT group (9%) than in the EC group (3%).

A higher proportion of MMMT patients (76%) presented at advanced stages (II–IV) than EC patients (58.3%). Most EC patients had histologically proven endometrioid adenocarcinoma subtypes (86%), while the remaining 14% had more aggressive subtypes, papillary serous, clear cell, undifferentiated, and high-grade mixed types. Appendix A includes a detailed description of patient demographic and tumor staging information.

### 3.2. Descriptive Statistical Analysis

The descriptive statistical analysis was performed to summarize first-order parameters and volume for both the endometrial carcinoma (EC) and malignant mixed Müllerian tumor (MMMT) groups (Table 1). Skewness: The mean value for EC was 0.49 (SD: 0.57), and for MMMT, it was 0.22 (SD: 0.47). The median value for EC was 0.55, and MMMTs’ was 0.27. Volume: The mean value for EC was 5.5 cm^3^ (SD: 6532.11); for MMMT, it was 13.64 cm^3^ (SD: 16,182.72). The median value for EC was 3.22 cm^3^ and MMMT was 7.01 cm^3^. The analysis concluded that a higher skewness value and a lower tumor volume value were associated with EC.

### 3.3. Univariate Analysis of AUROC for First-Order, Volume, and GLCM Features

The univariate analysis of the AUC ROC for the first-order, volume, and GLCM features was performed to predict EC status. The results are presented in Appendix A. The top predictor was volume, with an AUC of 0.71, a lower limit (LL) of 0.57, an upper limit (UL) of 0.84, and a *p*-value of 0.006. The AUC’s LL > 0.5 indicates a significant predictor, while an AUC of 0.5 represents a random model. Other predictors with significant AUC values include various ranges and angular variances of cluster shade, sum of squares (variance), sum average, and information measure of correlation. These results suggest that the univariate analysis of the AUC ROC for the first-order, volume, and GLCM features can help differentiate between MMMT and EC.

### 3.4. Multivariate Analysis for First-Order, Volume, and GLCM Features at Baseline

We utilized multivariate analysis to construct a more robust radiomic model. Employing a final elastic net model and adopting a 5-fold cross-validation approach, we observed an average AUC of 0.685, accompanied by a standard error of 0.055. This iterative methodology, involving data splitting, model fitting, and validation, further substantiated the potential of radiomics in stratifying EC and MMMT. The culmination of this analysis, utilizing the entire dataset for model prediction, yielded an AUC of 0.877 (0.79~0.96).

A multivariate analysis was conducted on baseline first-order, volume, and gray-level co-occurrence matrix (GLCM) features. The obtained coefficients of elastic net regularization are presented in Table 2. Among the first-order statistical features, skewness demonstrates a coefficient of elastic net regularization for the standardized data of 0.471, indicating a negative association with the event. This suggests that an increase in skewness might lead to a decrease in the odds of the event. Notably, the ‘8 Range of Sum average’ feature holds a coefficient of 0.266, implying a similar negative correlation. Other first-order features such as ‘8 Range of Sum of squares Variance’ (coefficient: 0.242) and ‘8 Range of Sum entropy’ (coefficient: 0.209) also exhibit negative associations. The ‘Volume’ feature is associated with a coefficient of 0.176, signifying a weaker negative relationship. In the context of GLCM features, ‘256 Angular Variance of Homogeneity’ (coefficient: 0.139) and ‘256 Angular Variance of Entropy’ (coefficient: 0.089) indicate less pronounced negative associations.

The provided table presents the coefficients for various radiomic features and the outcome to represent EC. A coefficient indicates the strength of association between a feature and the outcome of interest. For instance,

Skewness: A one-unit increase in skewness corresponds to a roughly 52.89% decrease in the odds of the outcome, indicating a link between lower skewness values and the outcome.Various features, such as the range of sum average, range of a sum of squares/variance, and range of sum entropy (8 bins), are associated with odds decreases of around 73.40%, 75.77%, and 79.07%, respectively.A higher volume leads to an approximately 82.37% decrease in the odds of the outcome.Angular variance of homogeneity (256 bins) is linked to an odds reduction of about 86.12% and angular variance of entropy (256 bins) to a decrease of about 91.08%.A higher average of homogeneity (256 bins) and an average of information measure of correlation (32 bins) are associated with odds decreases of approximately 92.78% and 95.49%, respectively.Similar odds reductions are seen for other features, such as the range of sum of squares/variance (16 bins) and average of cluster prominence (8 bins).Higher values of the average cluster shade (64 bins) and angular variance of the sum average (8 and 16 bins) correspond to significant odds decreases of up to 99.97%.

These coefficients provide insights into how changes in the radiomic features are associated with changes in the odds of the outcome, suggesting which features might be more significant to the studied outcome.

Interestingly, some features present near-zero or very low coefficients, implying a minimal impact on the odds of the event. For instance, ‘8 Average of Cluster Prominence’ (coefficient: 0.005) and ‘8 Angular Variance of Sum average’ (coefficient: 0.003) have minimal relevance. Moreover, the ‘64 Average of Cluster Shade’ (coefficient: 0.0008) and ‘16 Angular Variance of Sum average’ (coefficient: 0.0003) demonstrate weak associations with the event. These coefficients derived from elastic net regularization provide valuable insights into the varying degrees of influence that different features have on the odds of the event. The results aid in understanding the relative importance of individual components in predicting the outcome and contribute to feature selection and model refinement in the context of the analyzed dataset.

### 3.5. Overall Survival Prognosis

The survival analysis reported notable disparities in the overall survival rates between these malignancies (Table 3). The overall survival was significantly longer (*p* < 0.022) in patients with EC versus those with MMMT (Figure 2).

### 3.6. Multivariate Cox Regression

An independent association between radiomic features and overall survival within the EC and MMMT subgroups was analyzed using a multivariate Cox regression model. The 256 Angular Variance of Energy was an independent predictor of overall survival, particularly within the EC versus MMMT grouping *p* < 0.004 (Table 4).

## 4. Discussion

Our study provides valuable insights into the application of MRI-based radiomics in distinguishing between malignant mixed Müllerian tumors (MMMTs) and endometrial carcinoma (EC). Our findings reveal distinctive radiomic features that can aid in this differentiation. Specifically, we observed that EC exhibited higher skewness values and lower tumor volumes than MMMTs, which agrees with emerging evidence highlighting the potential of radiomics to characterize these malignancies.

Our results align with recent research in the field. A similar trend of elevated skewness values was observed in endometrial carcinoma cases, supporting our finding and emphasizing the consistent role of skewness in reflecting tumor heterogeneity across different cancer types [10,11,12,13,14,15]. Additionally, recent work by Juan et al. underscored the significance of skewness variations in stratifying breast cancer patients based on Ki-67 expression, further supporting our finding of skewness as a marker of prognostic value [16]. Our study’s integration of skewness and tumor volume into a predictive model with a high AUC aligns with the direction of radiomics research. Recent studies by Qian et al. and Jajodia et al. demonstrated the utility of similar integrated models in enhancing diagnostic accuracy for various cancer types, including ovarian and cervical cancers [17,18]. This collective body of work underscores the potential of combining radiomic features to achieve robust predictive models in gynecological oncology.

While our study is pioneering in directly comparing radiomic features of EC with MMMT, recent investigations have explored related avenues—for instance, a study by Zheng et al. [19] evaluated the diagnostic performance of radiomics in distinguishing EC subtypes. Additionally, the study by Fasmer et al. [20] comparing single-slice versus whole-tumor-derived radiomic signatures aligns with our emphasis on considering the entire tumor volume for improved predictive performance. Our study’s relevance extends beyond diagnosis, mirroring broader trends in radiomics research. The success of radiomic–clinical models in differentiating endometrial carcinoma from hyperplasia, as demonstrated by Zhang et al., [21] underscores the clinical applicability of radiomics beyond distinguishing malignancies.

Recent advancements in medical imaging and data analysis have highlighted the crucial role of tumor volume in understanding and managing various cancers. Tumor volume has been used as a prognostic tool in multiple malignancies, including endometrial carcinoma. Studies have shown that tumor volume can effectively distinguish high-grade and low-grade endometrioid adenocarcinomas, with a sensitivity of 88% and specificity of 89% [22]. Tumor volumetry has been used in the initial staging of endometrial cancer and has shown correlations with deep myometrial invasion, tumor grade, and lymphovascular invasion [23,24]. Preoperative tumor size determined using magnetic resonance imaging (MRI) has also been associated with lymph node metastases and survival [25]. Furthermore, tumor volume on preoperative MRI has been found to correlate with poor prognostic factors, making it a valuable biomarker for managing endometrial carcinoma [26].

Our study suggested the 256 angular variance of energy was significantly associated with OS and was independent of the pathological diagnosis of EC vs. MMMT. Angular second-order statistics have been used to classify magnetic resonance brain images and provide quantitative information about the internal structure of tissues and lesions, which can be helpful in medical diagnosis and treatment planning [27]. In some studies, texture features derived from MRI have been associated with patient survival and outcomes in various conditions, such as glioblastoma [28]. In abdominal MRI, texture analysis has been applied to multiple conditions, such as liver fibrosis [29], hepatocellular carcinoma [30], and Crohn’s disease [31]. However, the specific role of angular second-order statistics as a significant predictor in abdominal MRI depends on the context and the condition being studied. For example, in a study on liver fibrosis, noncontrast MRI scans with texture analysis were viable for classifying the early stages of liver fibrosis, exhibiting excellent performance [29]. In another study on hepatocellular carcinoma [30], texture analysis was performed on retrospective CT/MRI images, and the results suggested that texture features could help predict the progression of the disease. In the case of Crohn’s disease [31], texture analysis parameters of contrast-enhanced MRI were found to differ according to the presence of histological markers of hypoxia and angiogenesis.

## 5. Limitations

This study acknowledges that its sample size is relatively small, which could limit the generalizability of the findings. The rarity of MMMT contributes to this limitation. A larger sample size might provide more robust and representative results. The study design is retrospective, which can introduce biases and limit the control over data collection and variables, potentially affecting the accuracy and reliability of the results. The study data are collected from a single institution, which could lead to selection bias and limit the diversity of patient populations and imaging protocols. Multi-center studies might provide a more comprehensive view of the problem. MRI studies were conducted on machines with varying parameters, including other coils and field strengths. This variability in hardware might introduce confounding factors that could affect the radiomic features extracted. The tumor was segmented manually, introducing subjectivity and potential variability in delineating the regions of interest. Automated or semi-automated segmentation methods enhance accuracy and reproducibility. The patient distribution between the EC and MMMT groups is unbalanced, with fewer EC patients. This could affect the statistical analyses and interpretations. Radiomic studies often include highly correlated features, which may provide redundant information and affect the performance of predictive models. The lack of external validation results limits the generalizability and reproducibility of the findings in different patient populations and imaging settings.

## 6. Future Directions and Clinical Implications

The application of radiomics analysis in endometrial carcinoma and malignant mixed Müllerian tumors has shown promising results in differentiating these two types of tumors and predicting clinical outcomes. However, further research is needed to validate these findings in larger cohorts and explore the potential clinical implications of these radiomic features in diagnosing and managing patients with MMMT and EC. Additionally, integrating radiomic features with other clinical, pathological, and molecular data may provide a more comprehensive understanding of tumor biology and improve the accuracy of risk stratification and treatment planning.

## 7. Conclusions

This study demonstrates an initial step to investigate MRI-based radiomics to differentiate between MMMT and EC. The radiomic analysis revealed specific features such as volume and particular gray-level co-occurrence matrix (GLCM) features that distinguish these malignancies. Increased skewness and decreased volume were associated with EC and angular variance was an independent factor for OS. This study also highlights the importance of analyzing radiomic features in predicting patient survival outcomes. The findings support the idea that radiomics may offer valuable insights into distinguishing between MMMT and EC and potentially aid clinical decision-making and patient management. Given the small sample size further research is needed with more extensive and diverse datasets, prospective designs, and standardized imaging protocols to validate and refine the proposed radiomic approach for differentiating and for the prognosis of these gynecologic malignancies. Integrating radiomics with other clinical and pathological indicators may offer a more comprehensive understanding of the diseases and potentially enhance patient care.

## Figures and Tables

**Figure 1 cancers-16-02647-f001:**
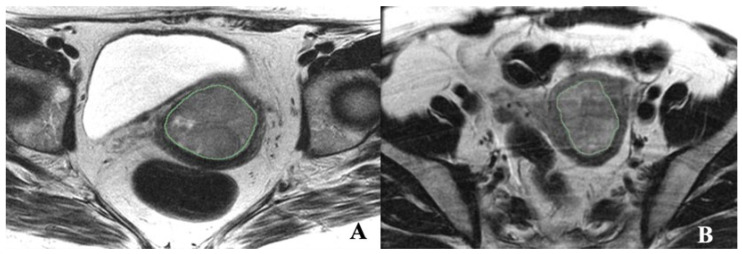
Illustrations of texture features analysis. (**A**) Axial T2-weighted MRI image of a 45-year-old female with EC. (**B**) Axial T- weighted MRI image of a 58-year-old female with MMMT.

**Figure 2 cancers-16-02647-f002:**
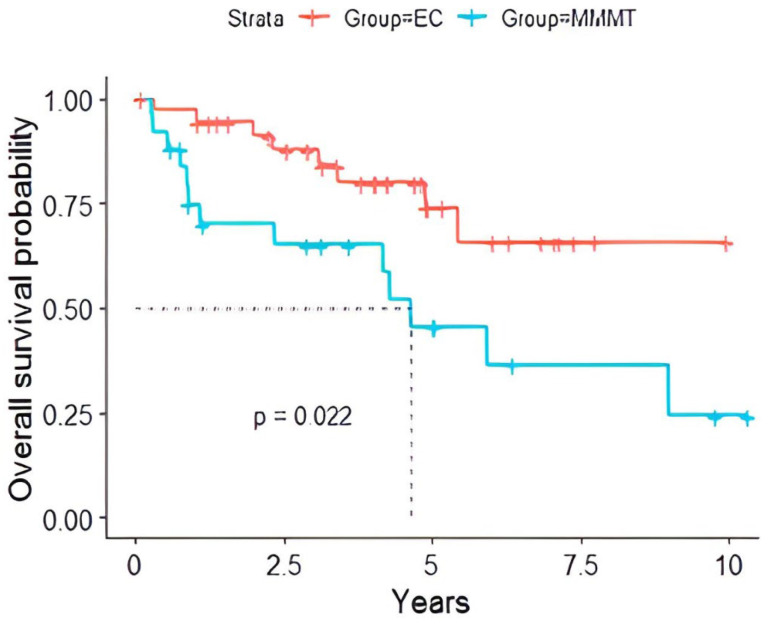
Overall survival probability of EC versus MMMT patients with EC showed a significantly prolonged survival (*p* = 0.022) compared to those with MMMT. The dotted line represents the median survival time.

**Table 1 cancers-16-02647-t001:** Descriptive statistical analysis to summarize first-order parameters and volume for EC and MMMT.

	EC (N = 36)	MMMT (N = 25)	Total (N = 61)	*p* Value
**Minimum**				0.348
N	36	25	61	
Mean (SD)	131.14 (167.63)	79.90 (64.03)	110.14 (136.66)	
Median (Range)	69.00 (4.00, 872.00)	64.00 (9.00, 246.00)	67.00 (4.00, 872.00)	
**Maximum**				0.628
N	36	25	61	
Mean (SD)	411.81 (283.91)	402.52 (276.26)	408.00 (278.51)	
Median (Range)	313.50 (115.00, 1496.00)	291.00 (112.00, 1077.00)	311.00 (112.00, 1496.00)	
**Mean**				0.918
N	36	25	61	
Mean (SD)	244.21 (206.65)	225.47 (152.38)	236.53 (185.16)	
Median (Range)	167.33 (58.99, 1075.43)	179.69 (55.22, 579.68)	173.40 (55.22, 1075.43)	
**Standard Deviation**				0.587
N	36	25	61	
Mean (SD)	45.14 (30.75)	46.46 (38.07)	45.68 (33.64)	
Median (Range)	35.93 (10.66, 159.46)	27.84 (12.94, 129.67)	35.40 (10.66, 159.46)	
**Percentile 1**				0.730
N	36	25	61	
Mean (SD)	159.11 (171.05)	125.02 (84.76)	145.14 (142.22)	
Median (Range)	96.38 (30.24, 915.58)	109.12 (30.00, 306.55)	106.00 (30.00, 915.58)	
**Percentile 5**				0.849
N	36	25	61	
Mean (SD)	178.66 (174.55)	152.41 (100.83)	167.90 (148.35)	
Median (Range)	124.50 (42.00, 938.00)	128.00 (36.00, 374.92)	126.00 (36.00, 938.00)	
**Percentile 95**				0.730
N	36	25	61	
Mean (SD)	323.19 (247.80)	304.04 (212.38)	315.34 (232.27)	
Median (Range)	253.40 (77.00, 1251.20)	226.25 (78.00, 771.86)	253.00 (77.00, 1251.20)	
**Percentile 99**				0.603
N	36	25	61	
Mean (SD)	364.11 (265.18)	341.80 (241.91)	354.97 (254.07)	
Median (Range)	280.74 (87.00, 1342.26)	242.25 (90.00, 916.29)	277.40 (87.00, 1342.26)	
**Skewness**				0.045
N	36	25	61	
Mean (SD)	0.49 (0.57)	0.22 (0.47)	0.38 (0.55)	
Median (Range)	0.55 (−1.07, 2.18)	0.27 (−1.00, 0.88)	0.45 (−1.07, 2.18)	
**Kurtosis**				0.557
N	36	25	61	
Mean (SD)	3.81 (1.41)	3.55 (0.95)	3.71 (1.24)	
Median (Range)	3.58 (1.90, 9.24)	3.42 (2.30, 6.92)	3.45 (1.90, 9.24)	
**Volume**				0.007
N	36	25	61	
Mean (SD)	5541.92 (6532.11)	13,646.22 (16,182.72)	8863.35 (12,074.47)	
Median (Range)	3215.29 (280.96, 31,591.33)	7011.87 (691.41, 64,373.78)	5603.25 (280.96, 64,373.78)	

Endometrial carcinoma (EC); malignant mixed Müllerian tumor (MMMT). Higher skewness and lower tumor volume was associated with EC.

**Table 2 cancers-16-02647-t002:** Final elastic net regularization model.

Feature	Coefficient
Skewness	0.47
8 Range of Sum average	0.27
8 Range of Sum of squares Variance	0.24
8 Range of Sum entropy	0.21
Volume	0.18
16 Range of Sum average	0.16
256 Angular Variance of Homogeneity	0.14
256 Angular Variance of Entropy	0.089
256 Average of Homogeneity	0.072
32 Average of Information measure of correlation 1	0.045
16 Range of Sum of squares Variance	0.040
8 Average of Cluster Prominence	0.0055
8 Angular Variance of Sum average	0.0033
64 Average of Cluster Shade	0.00080
16 Angular Variance of Sum average	0.00030

**Table 3 cancers-16-02647-t003:** Overall survival of those with EC versus MMMT.

Time (Years)	EC			MMMT		
	Survival (%)	L CI (%)	U CI (%)	Survival (%)	L CI (%)	U CI (%)
2	94.3%	86.9%	100%	70.1%	53.7%	91.5%
5	73.9%	58.1%	93.9%	45.6%	27.5%	75.6%
7	65.6%	47.1%	91.6%	36.5%	18.7%	71.2%

Confidence interval (CI); endometrial carcinoma (EC); malignant mixed Müllerian tumor (MMMT).

**Table 4 cancers-16-02647-t004:** Multivariate Cox regression analysis for odds ratio using texture features.

	HR	CI Lower HR	CI Upper HR	*p* Value
256 Angular Variance of Energy	1.081	1.025	1.140	0.004
Group (MMMT as reference)	2.297	0.925	5.702	0.073

256 Angular variance of the energy was significantly associated with the odds ratio (OS) and was independent of the pathological diagnosis of EC vs. MMMT.

## Data Availability

All data are available upon request.

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
