# Peer review of "MRI Radiomics Data Analysis for Differentiation between Malignant Mixed Müllerian Tumors and Endometrial Carcinoma"

_cancers, 2024, doi:10.3390/cancers16152647_

Round 1

Reviewer 1 Report

Comments and Suggestions for Authors I would like to congratulate the researchers on carrying out very interesting research on the issues  related specific radiomic features that could differentiate these malignancies Müllerian tumors and endometrial carcinoma. The aim of this research was to presents a comprehensive analysis of radiomic features extracted from MRI images of patients diagnosed with EC and MMMT. The authors  highlights the importance of analyzing radiomic features in predicting patient survival outcomes and potentially aid clinical decision-making and patient management. It is an interesting and important article. The differentiation between these malignancies holds paramount clinical significance due to their divergent treatment approaches, prognostic implications, and therapeutic outcomes. >From the scientific point of view the study is correctly designed. The article contains all the elements, the individual sections are well developed. The results are presented clearly and legibly in the form of tables and figures. The subject matter of the manuscript  corresponds to the scope of research proposed by the Journal. I recommend this paper for publicaton in Cancers.   There are some comments for this interesting manuscript:

  • The study is correctly designed.
  • The results provide an advancement of the current knowledge.
  • The cited references are mostly relevant recent publications.
  • The study has some limitations: the small sample size, retrospective nature, and potential biases. The authors indicate areas that should be further analyzed and  which it is worth to continue further experiments.

  There are some comments that could help to improve interesting manuscript: 1. The text requires editorial correction, 2. Please correct  Table 3: missing decimal point? 3. References: please complete all the references in accordance with the Instructions for the authors.  I recommend this paper for publication in Cancers after minor revision.    Thank you for the opportunity to review this manuscript.

Reviewer 2 Report

Comments and Suggestions for Authors

This is a pilot retrospectve cohort study to show differences in Radiomics between EC and Carcinosarcomas

I have these comments

Please add demographics and stages (grades of EC as well) of the patients and tumours included

Not sure how many G3 EC were included

Please put inclusion and exclusion criteria as separate sections

Please show the examples of radiomics in MRI as not all readers are familiar with it. It is good to see the quantifying radiomics in action

Comments on the Quality of English Language

This is a pilot retrospectve cohort study to show differences in Radiomics between EC and Carcinosarcomas

I have these comments

Please add demographics and stages (grades of EC as well) of the patients and tumours included

Not sure how many G3 EC were included

Please put inclusion and exclusion criteria as separate sections

Please show the examples of radiomics in MRI as not all readers are familiar with it. It is good to see the quantifying radiomics in action

Reviewer 3 Report

Comments and Suggestions for Authors

1. Table 3 7 years survival, the number of % lost "." .

2. Clinical doctors have applied relatively little in this area, and there is a lack of discussion on why the survival period is affected.

Round 2

Reviewer 2 Report

Comments and Suggestions for Authors

The Authors responded and corrected as per comments